# Harnessing Mitochondrial Stress for Health and Disease: Opportunities and Challenges

**DOI:** 10.3390/biology13060394

**Published:** 2024-05-29

**Authors:** Yujia Sun, Linlu Jin, Yixue Qin, Zhi Ouyang, Jian Zhong, Ye Zeng

**Affiliations:** Institute of Biomedical Engineering, West China School of Basic Medical Sciences & Forensic Medicine, Sichuan University, Chengdu 610041, China; 2022151610140@stu.scu.edu.cn (Y.S.); 2023224010042@stu.scu.edu.cn (L.J.); 2023224010044@stu.scu.edu.cn (Y.Q.); ouyangzhi@stu.scu.edu.cn (Z.O.); mailofz@stu.scu.edu.cn (J.Z.)

**Keywords:** mitohormesis, mitochondrial dysfunction, reactive oxygen species, cellular metabolism

## Abstract

**Simple Summary:**

Mitochondria could respond adaptively to mild stress, a process known as mitohormesis. This review introduces the various factors that can induce mitohormesis, including hydrogen sulfide (H_2_S), physical stimulation, exercise, reactive oxygen species (ROS), calcium, unfolded mitochondrial proteins (UPRmt), and integrated stress response (ISR). These factors regulate mitochondrial hormetic stimulation through mechanisms such as mitochondrial quality control (MQC) and mitophagy, which are essential for maintaining mitochondrial homeostasis. The modulation of mitochondrial stress through mitohormesis holds great promise for improving human health and treating a range of diseases. The review emphasizes the need for further exploration of the underlying mechanisms and the development of targeted therapeutic strategies that harness the power of mitochondria to promote longevity and combat age-related diseases.

**Abstract:**

Mitochondria, essential organelles orchestrating cellular metabolism, have emerged as central players in various disease pathologies. Recent research has shed light on mitohormesis, a concept proposing an adaptive response of mitochondria to minor disturbances in homeostasis, offering novel therapeutic avenues for mitochondria-related diseases. This comprehensive review explores the concept of mitohormesis, elucidating its induction mechanisms and occurrence. Intracellular molecules like reactive oxygen species (ROS), calcium, mitochondrial unfolded proteins (UPRmt), and integrated stress response (ISR), along with external factors such as hydrogen sulfide (H_2_S), physical stimuli, and exercise, play pivotal roles in regulating mitohormesis. Based on the available evidence, we elucidate how mitohormesis maintains mitochondrial homeostasis through mechanisms like mitochondrial quality control and mitophagy. Furthermore, the regulatory role of mitohormesis in mitochondria-related diseases is discussed. By envisioning future applications, this review underscores the significance of mitohormesis as a potential therapeutic target, paving the way for innovative interventions in disease management.

## 1. Introduction

Mitochondria, multifunctional organelles crucial for cellular activities, regulate various processes such as calcium homeostasis, antioxidant response, synthesis of metabolic intermediates, inflammation, cell death, proliferation, and signal transduction. Moreover, they serve as the primary energy regenerators sustaining life activities. Dysfunction in mitochondria correlates strongly with numerous human diseases, including neurodegenerative disorders, obesity, diabetes, cancer, and aging. Therefore, mitigating mitochondrial dysfunction is important in combating the onset and progression of these diseases and aging. Perturbations in mitochondrial function can impede the mitochondrial quality control (MQC) system, eliciting cellular responses to restore metabolism and homeostasis. This MQC system operates across various levels of cellular organization [1].

At the molecular level, chaperones and mitochondrial proteases prevent protein misfolding and aggregation within mitochondria. At the organelle level, dysfunctional mitochondria undergo recognition and degradation via autophagy including non-selective autophagy and mitophagy, facilitating specific recognition and clearance of damaged mitochondria.

Reactive oxygen species (ROS), byproducts of mitochondrial oxidative phosphorylation, participate in numerous physiological processes and serve as an important cellular messenger in signal transduction pathways. However, excessive ROS production can lead to severe oxidative stress and damage cells. Over a decade ago, the concept of a general toxicant excitation process was proposed, wherein mild mitochondrial stress triggers cellular changes that enhance resistance to subsequent injury [2]. Recently, researchers have focused on the regulatory effects of ROS on mitochondrial function, especially mitohormesis. Mitohormesis involves mild disturbances in mitochondrial homeostasis that coordinate communication between mitochondria and nucleus, thus reducing cellular susceptibility to external stimuli [3].

In the present work, we review the advances on mitohormesis, emphasizing how mitochondria sense external stimuli such as mechanical forces, oxidative stress recovery, hypoxia, hydrogen sulfide (H_2_S), and intracellular signaling such as ROS, calcium (Ca^2+^), and the mitochondrial unfolded protein response (UPRmt). The recent advancements in understanding how these signals induce mitochondrial stress are highlighted to elucidate the molecular mechanisms of mitohormesis in human disease and aging.

## 2. Overview of Mitohormesis

Toxicant excitation refers to a phenomenon where cells and organisms undergo adaptive changes upon exposure to low doses of potentially harmful stressors. This adaptive response helps maintain physiological balance, enabling better tolerance to subsequent stressors, while exposure to high doses beyond a threshold can result in cellular damage [3].

The concept of mitohormesis was initially proposed in 2006 [4]. Various stimuli can induce mitohormesis, including elevated levels of mitochondria ROS, increased metabolic products such as ATP, calcium ions, and nicotinamide adenine dinucleotide (NAD^+^), and disturbances in mitochondrial protein homeostasis. These stimuli induce mitochondria to engage in activities conducive to cell survival, primarily involving alterations in the expression levels of endogenous antioxidants and mitophagy within a manageable range.

Mitochondrial respiratory efficiency and stress recovery capabilities represent adaptive responses to oxidative stress, primarily mediated by ROS, which serve as the principal signaling molecule regulating mitohormesis. Mild oxidative stress functions as a survival adaptation mechanism for mitochondria. However, excessive oxidative stress can lead to irreversible damage to both mitochondria and cells. Over time, cells adapt to tolerate higher levels of oxidative stress, resulting in beneficial outcomes. Therefore, mitohormesis denotes a continuous adaptive process induced by mitochondrial stress to shield cells from apoptosis and mitochondrial dysfunction [5,6].

Maintaining mitochondrial homeostasis is imperative for cell survival, with mitohormesis serving as the critical mechanism for its preservation. Mitohormesis encompasses stimulating mitochondrial biogenesis, enhancing mitochondrial dynamics, and bolstering mitochondrial antioxidant defenses. These mechanisms collectively ensure mitochondrial function and homeostasis.

Cells sense and respond to aberrant mitochondrial activities and general damages such as electron transport chain (ETC) inhibition, protein misfolding, and redox imbalances through the MQC system. Key regulators of these processes include transcriptional coactivators, notably peroxisome proliferator-activated receptor-γ coactivator 1-α (PGC-1α) [7]. The activation of PGC-1α leads to increased transcription factors nuclear respiratory factor 1 and 2 (NRF1/NRF2) expression, promoting the transcription of many mitochondrial genes that encode the mitochondrial respiratory chain complexes [8]. NRF1 and NRF2 also stimulate the synthesis of mitochondrial transcription factor A (TFAM), which then mediates the replication and transcription of mitochondrial DNA (mtDNA) and promotes the formation of new mitochondria [9,10]. Additionally, PGC-1α stimulates the transcription of antioxidant enzymes such as superoxide dismutase (SOD) and catalase, collaborating with antioxidant transcription factor nuclear factor erythroid 2-related factor 2 (NF2L2) to mitigate mitochondrial oxidative stress [11].

Mitochondrial integrity and metabolism undergo stringent regulation by DNA repair networks, antioxidants, and proteases or chaperones participating in the UPRmt [12]. Mitochondrial stress can transport signaling molecules to the nucleus, regulate the expression of nuclear genes, and trigger adaptive stress responses [13]. When nutrients are abundant, mitochondrial acetyl-coenzyme A (acetyl-CoA) is delivered to the nucleus for intracellular histone acetylation, promoting cell growth and proliferation [14]. Energy deficiency, including electron transport chain (ETC) dysfunction, reduces nucleoplasmic acetyl-CoA levels, limits histone acetylation, and induces autophagy [15]. Mitochondrial stress generates ROS. ROS can activate ROS sensors, triggering transcription-activating signaling cascades through redox-dependent modifications of proteins [16]. The ROS-activating signaling pathways lead to the expression of genes that mediate the adaptive response in the nucleus [17]. Ca^2+^ is released from the mitochondria, which activates transcription factors and improves calcium homeostasis and mitochondrial metabolism via Ca^2+^ sensitive protein kinases, phosphatases, and transcriptional suppressors [18]. Conversely, dysfunctional and damaged mitochondria necessitate removal via mitophagy, often accompanied by a reduction in mitochondrial membrane potential, facilitating the recruitment of E3 ubiquitin ligase Parkin to the mitochondrial outer membrane (OMM) by PTEN-induced kinase 1 (PINK1) [19]. This initiates the ubiquitination of OMM proteins, triggering mitophagy to remove abnormal mitochondria with protein misfolding or depolarization, thereby preserving mitochondrial homeostasis. However, in cases of irreversible mitochondrial damage, the overexpression of certain protein toxic signals activates additional protein quality control networks, including the UPRmt and mitophagy, but exacerbates mitochondrial dysfunction caused by deleterious mtDNA accumulation [20]. A diagram illustrates the excitatory effects of mitohormesis is shown in Figure 1.

## 3. Mechanism of Mitohormesis

### 3.1. Mechanism of Mitohormesis Induction

#### 3.1.1. Hydrogen Sulfide (H_2_S)

H_2_S, a colorless and volatile gas, exhibits both toxic and antioxidant properties [21,22]. Mammalian cells primarily synthesize H_2_S through four enzymatic pathways, including cysteine thiothione β synthase (CBS), cysteine thiothione γ lyase (CSE), 3-mercaptopyruvate thiotransferase (3MST) coupled with cysteine aminotransferase (CAT), and 3MST coupled with D-amino acid oxidase (DAO) [23]. H_2_S plays a pivotal role in inhibiting oxidative damage by preventing irreversible cysteine peroxidation [24] and activating the redox-sensitive transcription factor NF2L2 [25]. Upon oxidative stress, H_2_S donors that release H_2_S can S-sulfhydrate Kelch-like ECH-associated protein 1 (Keap1) at cysteine-151, leading to NF2L2’s dissociation from Keap1 and promoting NF2L2’s translocation from the cytoplasm to the nucleus to bind to the promoters containing the antioxidant-response element (ARE) sequence and subsequently promote the expression of cellular antioxidant and defense proteins [26].

As an endogenous antioxidant, H_2_S can react directly with and quench the superoxide anion (O_2_^−^) as well as other ROS [27]. H_2_S sulfhydrates p66Shc at cysteine-59 in the N-terminal CH2 domain. This modification disrupts the association between protein kinase C-βII (PKC_βII_) and p66Shc, thus leading to the inhibition of PKCβII-mediated p66Shc phosphorylation at Ser36. This, in turn, inhibits mitochondrial ROS production via suppressing p66Shc’s translocation to mitochondria [28]. Within the physiological H_2_S concentration, cytochrome C oxidase maintains its normal function, and the oxidation of sulfide contributes to the production of mitochondrial ATP [29]. In the heart, H_2_S increases the synthesis of reduced glutathione (GSH) and upregulates thioredoxin expression, mitigating oxidative stress [30]. Furthermore, H_2_S could protect the myocardium against ischemia/reperfusion (IR) injury through its anti-autophagy [31].

Endogenous H_2_S levels decline with age and are associated with various age-related diseases. Experiments have shown that H_2_S levels in the heart, liver, and kidney tissues of aging mice are significantly reduced [32]. H_2_S levels is abnormally low in the brains of Alzheimer’s patients [33]. Aging and age-related diseases are characterized by oxidative damage, which can be alleviated by the antioxidant properties of H_2_S [34].

Therefore, low levels of H_2_S can act as an antioxidant, clearing the oxidative stress-induced ROS production and accumulation, reducing the accumulation of lipid peroxidation products, increasing the generation of intracellular antioxidants, such as superoxide dismutase (SOD), catalase (CAT), glutathione peroxidase (GPx), and glutathione (GSH), and ultimately protecting cells and tissues from oxidative stress damage [23]. However, insufficient H_2_S production exacerbates oxidative stress, leading to cellular and tissue damage. H_2_S exhibits potential in promoting health and longevity and holds promise for combating aging and age-related diseases (Figure 2).

#### 3.1.2. Mechanical Stimulation and Oxidative Stress Recovery (OxSR)

The mechanical properties of the extracellular matrix (ECM) regulate cell differentiation, metabolism, and mitochondrial function, influencing various physiological and pathological processes [35,36]. Cells respond to ECM cues, including mechanical stimulation, in part through mitochondrial stress responses (including changes in mitochondrial morphology and HSF-1-dependent transcription), which can modulate cell metabolism and proliferation using elevated ROS, without activating ROS-mediated cell death [37]. This process is known as OxSR, which enhances cytoplasmic ROS buffering, protecting cells from oxidative damage. Mitochondria respond to mechanical and electromagnetic stimuli, contributing to cellular signaling pathways involved in oxidative stress regulation. These processes correlated with the expression levels of calcium-permeable typical short transient receptor potential channel 1 (TRPC1) [38]. Mechanical and electromagnetic cellular responses influence calcium signaling pathways, such as the storage-operated calcium entry (SOCE) pathway, which modulates mitogen-activated protein kinase (MAPK) signaling [39]. These responses vary depending on cell type and environmental conditions, impacting cellular survival and metabolism. TRPC1-mediated calcium entry activates the calcineurin/nuclear factor of activated T cells (NFAT) pathway [40,41], maintaining TRPC1 transcription [42], as well as supporting PGC-1α function and oxidative-muscle maintenance [43]. PGC-1α activity is regulated by post-translational phosphorylation by AMP-activated protein kinase (AMPK) (stimulated by a high AMP/ATP ratio) and NAD-dependent deacetylase Sirtuin-1 (Sirt1) deacetylation (stimulated by a high NAD^+^/NADH ratio) [44,45]. In healthy muscle cells, low levels of magnetically elevated oxidative stress induce beneficial adaptive developmental responses [39], whereas in metabolically inflammatory breast cancer cells, increased levels of magnetically induced oxidative stress disrupt cell survival but keep healthy cells unharmed [46]. However, the expression of TRPC1 differed in extremely low-frequency electromagnetic field amplitudes (1.5 mT [39] and 3 mT [47]) and exposure times (10 min [39] and 60 min [47]).

Manipulating integrin mechanosignaling and ECM stiffness influence mitochondrial function and ROS production, affecting cellular responses to oxidative stress [37]. Stiff extracellular matrix and hyperglycemia regulate the activity of SLC9A1 (Na^+^/H^+^ exchanger 1, NHE1) and Na^+^/Ca^2+^ exchanger (NCX) through Rho-associated protein kinase (ROCK), inducing mitochondrial calcium overload, ROS production, and heat shock factor 1 (HSF1) transcription, thereby promoting metabolic adaptation and oxidative stress resilience [37]. Altering ECM stiffness can activate ROS-dependent pathways, leading to NF2L2 activation, which modulates sensitivity to chemotherapy treatments [48] (Figure 3).

### 3.2. Intracellular Signaling Mechanisms of Mitohormesis

#### 3.2.1. Reactive Oxygen Species

Mitochondria serve as the primary producers of ATP and ROS, with ROS recognized as a fundamental signaling molecule crucial for normal homeostatic function and the principal regulator of mitohormesis. In stem cells, ROS plays an indispensable role in tissue homeostasis maintenance and regeneration [49].

Various stimuli such as exercise, metabolic disorder, hypoxia, and mechanical stress elicit acute increases in ROS levels, triggering compensatory physiological responses in human body. These responses enhance the body’s tolerance to similar stressors, thereby improving overall resilience [50]. Antioxidant supplementation can mitigate exercise-induced ROS responses [51,52]. Oxygen deprivation (0.5–3% O_2_) induces oxidative stress, impairing ATP production and increasing apoptosis [53,54]. Mitochondria counteract hypoxic injury by upregulating transcription factor hypoxia-inducible factor (HIF) expression, releasing ROS, sensing oxygen tension, and conserving ATP [55].

ROS induction through physical exercises also enhances insulin sensitivity in humans via mitohormesis [56]. ROS can induce autophagy, particularly mitophagy, in response to mild and transient oxidative stress in a DRP1-dependent manner, ensuring efficient resource utilization under conditions such as hypoxia and nutrient deprivation [57]. For example, amino acid starvation and growth factor deprivation rapidly induce ROS, inducing non-selective autophagy [58]. Nix promotes mitochondrial depolarization and the production of ROS induced by uncoupling agents such as carbonyl cyanide mechlorophenzone (CCCP), thereby inhibiting mTOR signaling and activating autophagy [59]. However, excessive ROS levels can overload MQC systems, leading to irreversible damage [57].

In skeletal muscle, ROS signaling facilitates adaptation to increased contractile activity (i.e., muscle movement) or prolonged periods of inactivity (e.g., immobilization). Moderate ROS elevation during exercise activates signaling pathways promoting cellular adaptation and protecting against future stressors. Prolonged high ROS production may, however, trigger the chronic activation of pathways promoting proteolysis and potential cell death [60]. Redox-sensitive molecules including redox-sensitive kinases, phosphatases, and transcription factors, such as nuclear factor kappa-light-chain-enhancer of activated B cells (NF-κB) and mitogen-activated protein kinase (MAPK) family members, are induced by ROS, modulating processes like muscle protein breakdown and loss of nuclei via myonuclear apoptosis [61,62,63]. Furthermore, energy deficiency and hypoxia induce signaling pathways associated with longevity, including transcription factors such as HIF, FOXO, NF-κB, NRF2, and p53, and protein kinases such as AMPK mTOR, ULK1, HDAC5, and SIRT1 [64]. This adaptive response protects cells from stress and promotes cellular metabolic adaptation.

Phosphatases are divided into two main classes (serine/threonine phosphatases and phosphotyrosine phosphatases). Serine/threonine phosphatases contain readily oxidized metal ions. Ca^2+^/calcineurin is involved in muscle hypertrophy and fiber phenotypic transformation by dephosphorylates NFAT, translocating NFAT to the nucleus and increasing the expression of genes associated with muscle hypertrophy [65,66,67,68]. Phosphotyrosine phosphatases (PTPs) contain conserved cysteine residues at their active sites, which are susceptible to oxidation, thereby inducing enzyme inactivation [69]. Phosphatidylinositol 3-phosphatase (PTEN) is extremely important in skeletal muscle, and active PTEN dephosphorylates phosphatidylinositol (3,4,5)-triphosphate, blocking many cellular signaling pathways, including the activation of Akt [70]. The NF-κB family is composed of five related transcription factors (p65, RelB, c-Rel, p52, and p50), all of which are expressed in skeletal muscle. These transcription factors share an N-terminal DNA-binding/dimerization domain, known as the Rel homology domain, through which they can form homo- and heterodimers [71]. In non-stressed cells, the nuclear localization sequence of NF-κB binds to the inhibitory IκB protein in the cytoplasm, preventing the dimerization of p50 and p65, thereby preventing its entry into the nucleus. However, increased cytoplasmic ROS levels can activate IκB kinase (IKK), leading to the phosphorylation of the IκB protein, initiating ubiquitination, the degradation of IκB by proteasome, the elimination of inhibition, and the release of the NF-κB complex, resulting in dimerization and nuclear translocation. Although ROS can promote the activation and subsequent gene expression of NF-κB, the DNA-binding activity of oxidized NF-κB is reduced, suggesting that ROS may inhibit the transcriptional activity of NF-κB [72].

#### 3.2.2. Mitochondrial Unfolded Protein Response (UPRmt) and Integrated Stress Response (ISR)

Mitochondrial protein quality control mechanisms, particularly the UPRmt, play crucial roles in maintaining cellular homeostasis [73]. The UPRmt is activated in response to various mitochondrial dysfunctions, including abnormal protein import, protein misfolding, and oxidative phosphorylation damage [74]. It triggers retrograde signaling from mitochondria to the nucleus, promoting pathways such as nuclear-encoded protein quality control, antioxidant mechanisms, mitochondrial biogenesis, and mitophagy to mitigate stress and restore mitochondrial function [74].

The UPRmt interacts synergistically with other mitochondrial quality control systems (such as oxidative stress homeostasis, mitochondrial biogenesis, and mitochondrial autophagy) and is initiated by chaperones or proteases attempting to correct or remove damaged proteins [75,76]. Mitochondrial import efficiency of the transcription factor ATFS-1 in C. elegans and potentially orthologous transcription factors in mammals (ATF4, ATF5, CHOP) play critical roles in the UPRmt activation, facilitating cellular adaptation and mitochondrial homeostasis [20]. The UPRmt activate transcription factors, such as C/EBP homologous protein (CHOP) and CCAAT enhancer binding protein β (C/EBPB), which in turn increase the expression of UPRmt genes. The activation of the UPRmt pathway enhances the adaptive stress response and helps maintain homeostasis by inducing the transcription of mitochondrial molecular chaperones and proteases [77].

In addition, the integrated stress response (ISR) is also thought to be an important factor in the UPRmt activation. The ISR is a retrograde signaling pathway that integrates different stimuli from many organelles, including mitochondria, endoplasmic reticulum (ER), and cytoplasm [78]. Under various stress conditions, including mtDNA defects, misfolded proteins, mitonuclear imbalance, and increased ROS, the ISR inhibits general protein translation while favoring the translation of a small subset of mRNAs by activating the α-subunit of eukaryotic initiation factor 2 (eIF2α) and translation of transcription factors such as ATF4, ATF5, and CHOP [79,80,81].

#### 3.2.3. Intracellular Ca^2+^

During endurance exercise, skeletal muscle experiences significant increases in mitochondrial Ca^2+^ levels during and for about 60 min, primarily regulated by the mitochondrial membrane potential [82,83]. Elevated ROS levels associated with an increased mitochondrial membrane potential further modulate Ca^2+^ dynamics [84]. Mitochondria-mediated changes in cytoplasmic Ca^2+^ serve as retrograde signals, driving cellular adaptation to the environment.

Intracellular Ca^2+^ elevation during muscle contraction regulates gene expression associated with movement, primarily mediated by Ca^2+^/calmodulin-dependent protein kinases (CaMK) [85]. Physical exercise activates CaMKs, stimulating PGC-1α expression, mitochondrial stress, and glucose transport [86,87,88]. Mitochondria in a high cytosolic calcium environment in turn may produce more ROS. The ROS-mediated inhibition of sarcoplasmic Ca^2+^-ATPase (SERCA1a) activity delays calcium reuptake into the sarcoplasmic reticulum, promoting an increase in cytosolic Ca^2+^ concentration [89].

Calcium, acting as an enzyme catalyst, facilitates several signaling pathways crucial for skeletal muscle adaptation to exercise, including calcineurin (protein phosphatase 2B) signaling involved in oxidative muscle phenotype (characterized by elevated resting calcium levels to 100–300 nM) consolidation [90,91,92].

## 4. Mitochondrial Stress and Disease

In recent years, the relationship between mitochondrial dysfunction and various diseases has garnered significant attention, particularly in the context of atherosclerosis, cancer, metabolic diseases, aging, and neurodegenerative diseases (Figure 4).

Mitochondrial dysfunction contributes to atherosclerosis by influencing plaque stability [93,94]. Early stages of atherosclerosis involve mtDNA damage, increased ROS production, and progressive respiratory chain dysfunction, leading to endothelial cell dysfunction and plaque vulnerability. Oxidative stress induces autophagy, which, when excessive, can exacerbate cell death and plaque instability [95,96]. Physiological shear stress maintains vascular endothelial cell mitochondrial homeostasis, while low shear stress-induced inflammation contributes to atherosclerosis progression. Loss of TET2 leads to the upregulated expression and activity of mitochondrial respiratory complex II subunit succinate dehydrogenase B (SDHB), disrupting mitochondrial homeostasis, resulting in ROS overproduction and endothelial cell dysfunction [97]. The ROS scavenger NAC can reduce SDHB overexpression and the TET2 shRNA-induced pyroptosis of vascular endothelial cells [98].

In cancer, apoptosis is a crucial mechanism for inhibiting tumor growth and eliminating cancer cells [99]. By increasing metabolic activity, apoptosis raises levels of ROS within cancer cells, effectively hindering tumor progression. However, excessive ROS can disrupt mitochondrial function, leading to mitochondrial membrane depolarization and the opening of the mitochondrial permeability transition pore (mPTP). This, in turn, activates the intrinsic apoptotic pathway, ultimately resulting in apoptosis and necrosis through the activation of caspases and RIPK1/RIPK3, respectively [100]. Targeting mitochondrial pathways may offer promising strategies for cancer treatment, low-grade mitochondrial stress and the adaptive stress response can mitigate the progression of diseases.

During obesity, characterized by excessive nutrient supply, mitochondrial ROS production increases, prompting the biogenesis of mitochondria and antioxidant defense [101]. Targeting mitochondrial pathways involved in metabolic regulation may hold promise for treating metabolic diseases and improving overall health.

Aging and neurodegenerative diseases are closely associated with mitochondrial dysfunction and increased production of ROS. Oxidative stress resulting from mitochondrial dysfunction is a key event in the pathogenesis of Parkinson’s disease (PD) [102]. In a variety of neurodegenerative diseases, the NF2L2 antioxidant pathway is disrupted, mitochondrial metabolism and ROS homeostasis are significantly impaired, and the level of PGC-1α is significantly reduced [103]. A mild inhibition of mitochondrial function has been shown to prolong lifespan. UPRmt genes are upregulated in Alzheimer’s disease (AD), PD, and other neurodegenerative diseases, suggesting a protective role against protein aggregation interfering with mitochondrial function [104,105]. Studies have shown that compounds like chrysanthemum and apigenin can inhibit mitochondrial respiration temporarily, induce early ROS production, and activate the AMPK/NRF-2/FOXO pathway, thereby promoting mitosis, enhancing oxidative stress capacity and cellular metabolic adaptation, and extending the lifespan of *C. elegans* [106]. These compounds delay aging and improve age-related diseases by targeting mitochondrial function, suggesting potential therapeutic strategies for mitigating neurodegeneration and aging-related decline.

Overall, current research only recognizes severe or irreversible mitochondrial damage in diseases such as atherosclerosis, cancer, metabolic diseases, aging, and neurodegenerative diseases, but the early stages of mitochondrial damage and its relationship to disease progression remain unclear. Focusing on early mitochondrial damage may allow for a more accurate assessment of suboptimal health conditions and holds promise for promoting the health by triggering mitohormesis. 

## 5. Summary and Perspectives

Mitochondrial stress, when maintained at low levels, has been shown to confer beneficial effects on overall health and longevity. Mitochondria, the powerhouses of the cell, play a crucial role in maintaining cellular homeostasis and responding to various internal and external stressors. However, excessive stress can lead to irreversible damage and contribute to the pathogenesis of various human diseases. Studies in model organisms like mice, yeast, worms, and flies have demonstrated that a mild inhibition of mitochondrial respiration can extend lifespan and delay aging [107]. In humans, drugs like metformin, which induce mild mitochondrial stress, are widely used for the treatment of type 2 diabetes, while compounds like resveratrol and NAD^+^ precursors hold promise for delaying aging and prolonging lifespan [108]. Similarly, physical exercise acts as a natural stimulus for inducing beneficial mitochondrial stress, although excessive exercise can have adverse effects on mitochondrial function and metabolic health. Deepening our understanding of the mechanisms underlying mitochondrial stress and harnessing this knowledge for drug development and targeted therapies could offer new avenues for treating a range of human diseases associated with obesity, cardiovascular disease, and aging.

Potential future perspectives include: (1) Mitochondrial stress modulation: Explore novel therapeutic strategies aimed at modulating mitochondrial stress levels to promote health and treat diseases. This may involve the development of drugs that selectively induce mild mitochondrial stress or enhance mitochondrial quality control mechanisms. (2) Precision medicine approaches: Develop personalized treatment regimens based on individual mitochondrial profiles and genetic susceptibilities. Precision medicine approaches could help tailor interventions to target specific mitochondrial defects in patients with mitochondrial-related diseases. (3) Lifestyle interventions: Investigate the role of lifestyle interventions, such as exercise and dietary modifications, in modulating mitochondrial stress and promoting overall health. Understanding how different lifestyle factors influence mitochondrial function could inform preventive strategies for disease management. (4) Biomarker development: Identify biomarkers of mitochondrial stress that can be used for early disease detection and monitoring treatment responses. Biomarkers could aid in the development of personalized treatment approaches and facilitate clinical decision-making. (5) Translational research: Translate findings from basic research on mitochondrial stress into clinical applications for disease prevention and treatment. Collaborations between basic scientists, clinicians, and pharmaceutical companies are essential for translating discoveries into effective therapeutic interventions.

In conclusion, the modulation of mitochondrial stress holds significant promise for improving human health and treating a range of diseases. By further exploring the mechanisms underlying mitochondrial stress and developing targeted therapeutic strategies, researchers can pave the way for innovative treatments that harness the power of mitochondria to promote longevity and combat age-related diseases.

## Figures and Tables

**Figure 1 biology-13-00394-f001:**
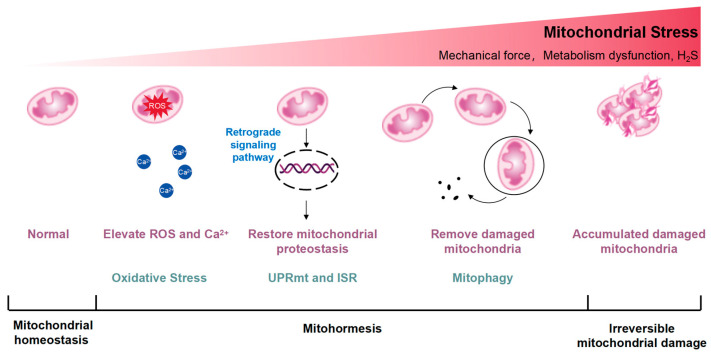
Conceptual diagram illustrating the excitatory effects of mitohormesis. Mitohormesis refers to an adaptive response that enhances cellular, tissue, or organismal vitality and health by inducing mild mitochondrial stress. The severity of the perceived stress determines the activation of different mitochondrial stress responses. The UPRmt acts as the initial defense mechanism against stress, influenced by reactive oxygen species (ROS) and intracellular calcium ion (Ca^2+^) increments, triggering retrograde signaling pathways from mitochondria to the nucleus to restore mitochondrial protein homeostasis. Mitochondrial stress can activate the integrated stress response (ISR), which is also thought to be an important factor in the activation of UPRmt. Additionally, mitochondria, being highly dynamic organelles, undergo coordinated cycles of fission and fusion, known as mitochondrial dynamics, to maintain their size, shape, and distribution. Mitochondrial dynamics are initiated by escalating stress levels, while autophagy, particularly mitophagy, occurs in response to the degradation of damaged mitochondria. As mitochondrial stress intensifies, irreparable damage may occur.

**Figure 2 biology-13-00394-f002:**
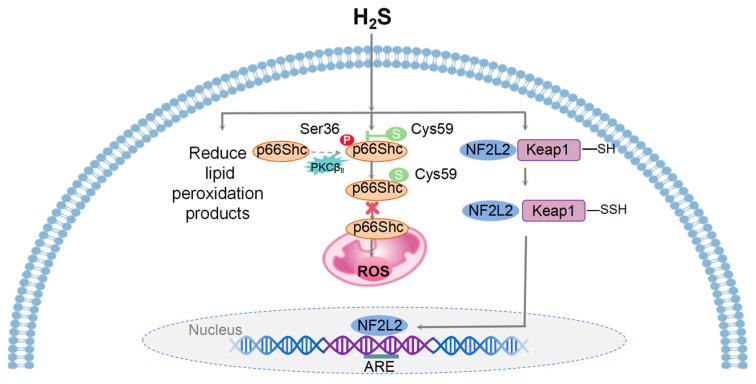
A model illustrating the mechanisms of mitohormesis induced by H_2_S. Physiological levels of H_2_S provide cellular and tissue protection against oxidative stress by ROS, reducing the accumulation of lipid peroxidation products, and enhancing the production of intracellular antioxidants. H_2_S inhibits mitochondrial ROS generation through the sulfhydration of the Cys-59 residue of p66Shc and its translocation to mitochondria. Upon oxidative stress, H_2_S facilitates the dissociation of NF2L2 from Keap1 and its translocation from the cytoplasm to the nucleus, where it binds to ARE and initiates the transcription of downstream target genes.

**Figure 3 biology-13-00394-f003:**
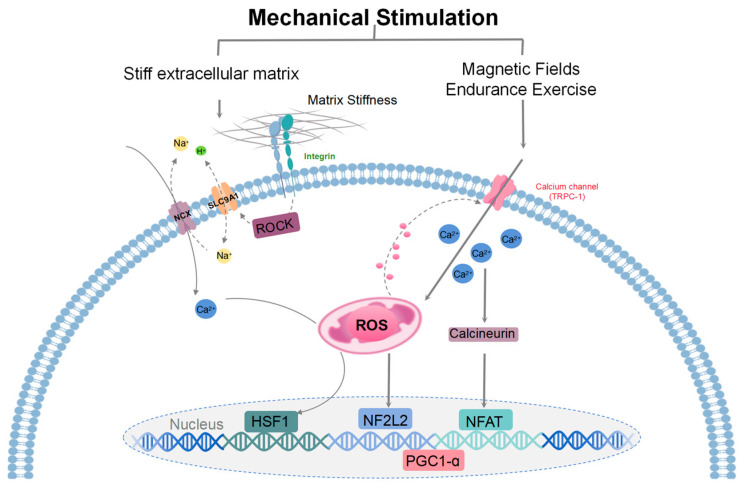
A model illustrating the mechanisms of mitohormesis induced by mechanical stimulation. Mechanosensing of a stiff extracellular matrix at adhesion sites activates ROCK to stimulate ion exchange by SLC9A1 and NCX. Subsequent mitochondrial calcium overload drives the production of ROS and stimulates the transcription of HSF1. HSF1-mediated mitochondrial stress suppresses the production of ROS by limiting mitochondrial respiration and opposes oxidant damage by promoting mitochondrial biogenesis/protein turnover and enhancing reducing equivalents (reduced glutathione/NADPH). Magnetic fields and endurance exercise signals are detected by transient receptor potential channels (TRPC), which regulate mitochondrial response and promote mitochondrial kinesis. During exercise training, metabolism increases sharply, repeatedly triggering the body’s compensatory physiological response, and ROS levels rise, which can be weakened by supplementation of antioxidants. These signaling pathways that activate ROS lead to gene expression that mediates the cellular adaptive responses. Key regulators of these processes include transcriptional coactivators, specifically PGC-1α. ROS regulate TRPC1, and TRPC1-mediated calcium entry participates in mechanical signaling by activating the calcineurin pathway to form various adaptive responses. Calmodulin-dependent calcineurin dephosphorylates and translocates NFAT to the nucleus. When ROS levels are elevated, NF2L2 binds to antioxidant reaction components in the nucleus and promotes the expression of cellular antioxidant and defense proteins.

**Figure 4 biology-13-00394-f004:**
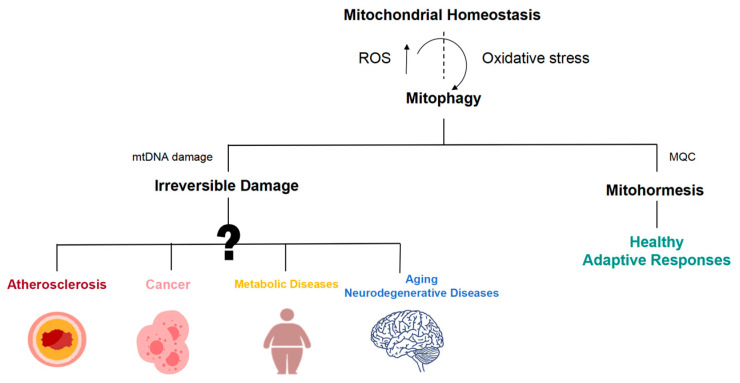
Mitohormetic responses are involved in the initiation and progression of a wide array of disease processes. Mitochondrial dysfunction is widely involved in the occurrence and development of atherosclerosis, cancer, aging, and neurodegeneration.

## Data Availability

Dataset available on request from the authors.

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
