# Peer review of "Harnessing Mitochondrial Stress for Health and Disease: Opportunities and Challenges"

_biology, 2024, doi:10.3390/biology13060394_

Round 1

Reviewer 1 Report

Comments and Suggestions for Authors

This review attempts to illustrate various mitochondrial quality control mechanisms, along with the role those mechanisms play in various types of disease.  The topic is important, and seems to comprehensively review various molecular mechanisms which is nice to have in one manuscript. However, the organization is a bit clunky, which makes the review hard to read.  

After reading through it three times, I'm still having trouble understanding the organization.  The overall scope is quite broad, but the content is so detailed that it reads as being too targeted and perhaps cherry picked.  Rather, each section could become its own manuscript, and that way the detailed information could be fleshed out, for a more comprehensive read.

The figures are helpful over all.  But, I would like to see Figure 2 larger (full-size page) with an expanded set of cell signaling mechanisms.  And each section could benefit from broader introductory and summary writing, prior to and after diving into the detail.  

Hopefully, this makes sense.  I'm in favor of publishing the review, but would like to see the writing and Figure 2 fleshed out to help increase its readability.

-The references are appropriate. 

-Overall, the topic is important, as mitochondria are vital, complex organelles that play a major role in cell health, cell maintenance, and cellular response to the environment. Small perturbations in mitochondrial homeostasis can have profound implications on cellular and tissue health, and may exacerbate vulnerability to disease.  The authors are attempting to cover a broad range of mitochondrial quality control topics, which could provide mitochondrial scientists with a important resource for understanding the extent of mitochondrial functional mechanisms.  Unfortunately, the review falls a bit short of this goal as each sub-topic is not covered in much depth.

-The major issue for me is that the authors are attempting to cover a broad range of topics regarding molecular mitochondrial quality control mechanisms but only providing a superficial explanation of those mechanisms citing various research articles.  As a reader, I'm not left with a strong understanding of any of the sub-topics because the extent of known mechanisms within each sub-topic are not included in full.  The window of coverage of the literature is too narrow. The schematics provided help a little, but should include more detail to reflect what is covered in the review. 

-I believe the review would benefit from narrowing the overall focus and expanding the detail of molecular mechanisms within each sub-topic.  The last section covering neurodegenerative disease, cancer and metabolic disease could be dropped entirely and replaced by a concluding paragraph that predicts the linkage between quality control mitochondrial mechanisms and health/disease.

Comments on the Quality of English Language

Some minor English editing (lines 41-33, line 30, line 49, line 82, for example) needed throughout.

Author Response

Dear Editors and Reviewer #1:

We would like to thank you and the reviewers for your comments on the improvement of our manuscript entitled Harnessing Mitochondrial Stress for Health and Disease: Opportunities and Challenges. The revisions are highlighted in the main text and a point-by-point response is provided below.

My Sincerely,

Ye Zeng

Sichuan University

Responds to the reviewer' s comments:

Reviewer #1: This review attempts to illustrate various mitochondrial quality control mechanisms, along with the role those mechanisms play in various types of disease. The topic is important, and seems to comprehensively review various molecular mechanisms which is nice to have in one manuscript. However, the organization is a bit clunky, which makes the review hard to read. After reading through it three times, I'm still having trouble understanding the organization. 

1.The overall scope is quite broad, but the content is so detailed that it reads as being too targeted and perhaps cherry picked.  Rather, each section could become its own manuscript, and that way the detailed information could be fleshed out, for a more comprehensive read. The figures are helpful over all.  But, I would like to see Figure 2 larger (full-size page) with an expanded set of cell signaling mechanisms. And each section could benefit from broader introductory and summary writing, prior to and after diving into the detail.  Hopefully, this makes sense.  I'm in favor of publishing the review, but would like to see the writing and Figure 2 fleshed out to help increase its readability.

Response to Q1 :

The reviewer's feedback is appreciated, and we understand the concern regarding the breadth and depth of the manuscript. In response to the suggestion that each section could become its own manuscript, we have taken a different approach to enhance the comprehensiveness and readability of the current review.

We have expanded the introductory and summary sections of each part to provide a broader context and a clearer overview of the topics discussed. This includes a more thorough explanation of the mechanisms and implications of mitohormesis induced by H2S and mechanical stimulation, as well as their relevance to cellular signaling and adaptation to oxidative stress.

For the figures, we have separated Figure 2 and now it occupies two half-pages. Additionally, we have expanded the set of cell signaling mechanisms depicted in the figure to provide a more detailed and comprehensive visual representation of the processes involved.

Mechanism of mitohormesis Iinduced by H2S is highlighted in Lines 158-168: “Upon oxidative stress, H2S donors that release H2S can S-sulfhydrate Kelch-like ECH-associated protein 1 (Keap1) at cysteine-151, leading to NF2L2 dissociation from Keap1 and promoting NF2L2 translocation from the cytoplasm to the nucleus to binds to the promoters containing antioxidant-response element (ARE) sequence and subsequently promote the expression of cellular antioxidant and defense proteins[26]. As an endogenous antioxidant medium, H2S can react directly with and quenches the superoxide anion (O2) as well as other ROS[27]. H2S sulfhydrates p66Shc at cysteine-59 in the N-terminal CH2 domain. This modification disrupts the association between protein kinase C-ßII (PKCßII) and p66Shc, and thus leading to the inhibition of PKCßII-mediated p66Shc phosphorylation at Ser36. This, in turn, inhibits mitochondrial ROS production via suppressing p66Shc translocation to mitochondria [28].” .

A model illustrating the mechanisms of mitohormesis induced by H2S is added in Lines 188-196:

Figure 2. A model illustrating the mechanisms of mitohormesis induced by H2S. Physiological levels of H2S provide cellular and tissue protection against oxidative stress by ROS, reducing the accumulation of lipid peroxidation products, and enhancing the production of intracellular antioxidants. H2S inhibits mitochondrial ROS generation through sulfhydration of Cys-59 residue. Upon oxidative stress, H2S facilitates the dissociation of NF2L2 from Keap1 and its translocation from the cytoplasm to the nucleus, where it binds to ARE and initiates transcription of downstream target genes.

These are revised in Section 3.1.1. now.

 Mechanism of mitohormesis Iinduced by mechanical stimulation is highlighted in Lines 206-234: “Mitochondria respond to mechanical and electromagnetic stimuli, contributing to cellular signaling pathways involved in oxidative stress regulation. These processes correlated with the expression levels of calcium permeable typical transient receptor potential Channel Type 1 (TRPC1)[38]. Mechanical and electromagnetic cellular responses influence calcium signaling pathways, such as the storage-operated calcium entry (SOCE) pathway, which modulates mitogen-activated protein kinase (MAPK) signaling[39]. These responses vary depending on cell type and environmental conditions, impacting cellular survival and metabolism. TRPC1-mediated calcium entry activates the calcineurin/NFAT pathway [40, 41], maintaining TRPC1 transcription[42], as well as supporting PGC-1α function and oxidative-muscle maintenance[43]. PGC-1α activity is regulated after translation by AMP-activated protein kinase (AMPK) phosphorylation (stimulated by a high AMP/ATP ratio) and NAD1-dependent deacetylase Sirtuin-1 (Sirt1) deacetylation (stimulated by a high NAD+/NADH ratio)[44, 45]. In healthy muscle cells, low levels of magnetically elevated oxidative stress induce beneficial adaptive developmental responses[39], whereas in metabolically inflammatory breast cancer cells, increased levels of magnetically induced oxidative stress disrupt cell survival but keep healthy cells unharmed[46]. However, the expression of TRPC1 differed in extremely low-frequency electromagnetic field amplitude (1.5mT[39] and 3mT[47], respectively) and exposure time (10 min[39] and 60 min[47], respectively).

Manipulating integrin mechanosignaling and ECM stiffness influences mitochondrial function and ROS production, affecting cellular responses to oxidative stress[37]. Stiff extracellular matrix and hyperglycaemia regulate the activity of SLC9A1 (Na+/H+ Exchanger 1 (NHE1)) and Na+/Ca2+ exchanger (NCX) through Rho-associated protein kinase (ROCK), inducing mitochondrial calcium overload, ROS production, and heat shock factor 1 (HSF1) transcription, thereby promoting metabolic adaptation and oxidative stress resilience[37]. Altering ECM stiffness can activate ROS-dependent pathways, leading to NF2L2 activation, which modulates sensitivity to chemotherapy treatments[48] (Figure 3).”.

A model illustrating the mechanisms of mitohormesis induced by mechanical stimulation is added in Lines 235-254:

Figure 3. A model illustrating the mechanisms of mitohormesis induced by mechanical stimulation. Mechanosensing of a stiff extracellular matrix at adhesions activates ROCK to stimulate ion exchange by SLC9A1 and NCX. Subsequent mitochondrial calcium overload drives the production of ROS and stimulates the transcription of HSF1. HSF1-mediated mitochondrial stress suppresses the production of ROS by limiting mitochondrial respiration, opposes oxidant damage by promoting mitochondrial biogenesis/protein turnover and enhancing reducing equivalents (reduced glutathione/NADPH). Magnetic fields and endurance exercise signals, detected by transient receptor Potential Specification (TRPC) channels, which regulate mitochondrial response and promote mitochondrial kinesis. During exercise training, metabolism increases sharply, repeatedly triggering the body's compensatory physiological response, and ROS levels rise, which can be weakened by supplementation of antioxidants. These signaling pathways that activate ROS lead to gene expression that mediates the cellular adaptive response. Key regulators of these processes include transcriptional coactivators, specifically the peroxisome proliferator activation receptor PGC-1α. ROS regulates TRPC1, and TRPC1-mediated calcium entry participates in mechanical signaling by activating the calcineurin pathway to form various adaptive responses. Calmodulin-dependent calcineurin dephosphorylates and translocates NFAT to the nucleus. When ROS levels are elevated, NF2L2 binds to antioxidant reaction components in the nucleus and promotes the expression of cellular antioxidant and defense proteins.

These are revised in Section 3.1.2. now.

  1. I believe the review would benefit from narrowing the overall focus and expanding the detail of molecular mechanisms within each sub-topic.  The last section covering neurodegenerative disease, cancer and metabolic disease could be dropped entirely and replaced by a concluding paragraph that predicts the linkage between quality control mitochondrial mechanisms and health/disease.

Response to Q2 : As suggested, we have narrowed the overall focus of the review and expanded the detail of molecular mechanisms within each sub-topic to provide a more in-depth analysis and a clearer understanding of the subject matter.

In response to the specific suggestion regarding the last section, we have removed the detailed discussion on mitochondria-related diseases and instead have included a concluding paragraph that predicts the linkage between quality control mitochondrial mechanisms and health/disease. This new section now serves to synthesize the information presented throughout the review and to provide a forward-looking perspective on the potential implications of mitohormesis for health and disease management.

The revised manuscript now includes a comprehensive and predictive discussion on how mitohormesis could play a role in promoting health and potentially mitigating the progression of diseases. We have retained the key points from the original section, such as the involvement of mitochondrial dysfunction in various diseases and the potential of targeting mitochondrial pathways for therapeutic interventions. However, the focus has been shifted to emphasize the preventative and health-promoting aspects of mitohormesis.

For details, we removed the section of mitochondria-related diseases and instead predicted the development of mitohormesis in promoting health. Lines 371-426: “In recent years, the relationship between mitochondrial dysfunction and various diseases has garnered significant attention, particularly in the context of atherosclerosis, cancer, metabolic diseases, aging, and neurodegenerative diseases (Figure 4).

Figure 4. Mitohormetic responses are involved in the initiation and progression of a wide array of disease processes. Mitochondrial dysfunction is widely involved in the occurrence and development of atherosclerosis, cancer, aging and neurodegeneration.

Mitochondrial dysfunction contributes to atherosclerosis by influencing plaque stability[93, 94]. Early stages of atherosclerosis involve mitochondrial DNA (mtDNA) damage, increased ROS production, and progressive respiratory chain dysfunction, leading to endothelial cell dysfunction and plaque vulnerability. Oxidative stress induces autophagy, which, when excessive, can exacerbate cell death and plaque instability[95, 96]. Physiological shear stress maintains vascular endothelial cell mitochondrial homeostasis, while low shear stress-induced inflammation contributes to atherosclerosis progression. Loss of TET2 leads to upregulated expression and activity of mitochondrial respiratory complex II subunit succinate dehydrogenase B (SDHB), disrupting mitochondrial homeostasis, resulting in ROS overproduction and endothelial cell dysfunction[97]. The ROS scavenger NAC can reduce SDHB overexpression and TET2 shRNA induced pyroptosis of vascular endothelial cells[98].

In cancer, apoptosis is a crucial mechanism for inhibiting tumor growth and eliminating cancer cells[99]. By increasing metabolic activity, apoptosis raises levels of ROS within cancer cells, effectively hindering tumor progression. However, excessive ROS can disrupt mitochondrial function, leading to mitochondrial membrane depolarization and the opening of the mitochondrial permeability transition pore (mPTP). This, in turn, activates the intrinsic apoptotic pathway, ultimately resulting in apoptosis and necrosis through the activation of caspases and RIPK1/RIPK3, respectively[100]. Targeting mitochondrial pathways may offer promising strategies for cancer treatment, low-grade mitochondrial stress and the adaptive stress response can mitigate the progression of diseases. During obesity, characterized by excessive nutrient supply, mitochondrial reactive oxygen species (ROS) production increases, prompting mitochondria to enhance biogenesis and antioxidant defense mechanisms[101]. Targeting mitochondrial pathways involved in metabolic regulation may hold promise for treating metabolic diseases and improving overall health.

Aging and neurodegenerative diseases are closely associated with mitochondrial dysfunction and increased production of ROS. Oxidative stress resulting from mitochondrial dysfunction is a key event in the pathogenesis of Parkinson's disease (PD)[102]. In a variety of neurodegenerative diseases, the NF2L2 antioxidant pathway is disrupted, mitochondrial metabolism and ROS homeostasis are significantly impaired, and the level of PGC-1α is significantly reduced[103]. Mild inhibition of mitochondrial function has been shown to prolong lifespan. UPRmt genes are upregulated in Alzheimer's disease (AD), PD, and other neurodegenerative diseases, suggesting a protective role against protein aggregation interfering with mitochondrial function[104, 105].Studies have shown that compounds like chrysanthemum and apigenin can inhibit mitochondrial respiration temporarily, induce early ROS production, and activate the AMPK/NRF-2/FOXO pathway, thereby promoting mitosis, enhancing oxidative stress capacity and cellular metabolic adaptation, and extending the lifespan of C. elegans[106]. These compounds delay aging and improve age-related diseases by targeting mitochondrial function, suggesting potential therapeutic strategies for mitigating neurodegeneration and aging-related decline.

Overall, current research only recognizes severe or irreversible mitochondrial damage in diseases such as atherosclerosis, cancer, metabolic diseases, aging, and neurodegenera-tive diseases, but the early stages of mitochondrial damage and its relationship to disease progression remain unclear. Focusing on early mitochondrial damage may allow for a more accurate assessment of suboptimal health conditions, and holds promising for pro-moting the health by triggering mitohormesis. ”.

These are added in the Section 4 now.

Reviewer 2 Report

Comments and Suggestions for Authors

In this review article, the authors overviewed the mechanisms by which mitochondrial stress induces both beneficial effects, known as mitohormesis, and harmfull effects that cause diseases and aging. The manuscript is well structured and covers important topics, however there are a number of unclear descriptions that need correction before publication.

Major points:

Since there are typing errors and use of unfamilier words, some of which is preferentially used by AI (e.g. delve, and paramount), the manuscript should be checked by native English speaker or such edditing service.

The meaning of the words "toxicant excitation process" (line 45), "mitochondiral excitatory effects" (line 71), and "mitochondrial excitation" (line 75) are unclear and they should be expressed differently or defined clearly.

Although the authors only mentioned "integrated stress response" (line 254), the ISR pathway should also be explaind sufficiently since DELE1-HRI mediated ISR is one of important retrograde pathway elicited by mitochondrial dysfunction. DELE1 is cleaved by OMA1, an upsteam of OPA1 (line 403), and FGF21 and GDF15 (line 399) are known as downstream target genes of ISR.

Minor point:

The words "synthesizes" (line 16), "paramount" (line 32), and "prompt" (line 66) should be changed appropriately.

Please check for grammatical errors. In line 30, "mitochondrial" may be "mitochondria". In line 105, "protein toxicity signals" may be "protein toxic signals" or "protein toxicities".

Since it is hard to distinguish NRF2 and Nrf2, that stand for Nuclear respiratory factor 2 and Nuclear factor erythroid 2-related factor 2, respectively, tha latter can be abbreviated as NFE2L2.

Please check all abbreviations are spelled out when they appeared in the first time in the abstract and in the main text, especially "mtDNA", "PGC-1α", "PTEN", "GSH", "MAPK", "SOD", and "HSF1".

Since full spell of some abbreviations are uncommon or wrong (e.g. "NF-κB", "SERCA", and "NFAT"), all abbreviations should also be checked using database such as GeneCards.

Common terms such as "NAD+" may not be spelled out.

Some abbreviations not appear in the later part may be removed such as "ELF EMF" (line 178), "EC" and "AS" (line 318).

In line 98, "cross-talk factors" is too broad in this context and should be corrected.

In line 120, the meaning of "pressure" is unclear.

Since TRPC1 (line 177) is known to contribute to SOCE (line 170), the explanation of this paragraph may be improved.

In line 194, "metabolism" may be "metabolic disorder" or "metabolic stress" because basal metabolism may not be stimulus.

In line 203, "mitotic effects" is unclear and not mentioned in the reference. This may be typing error of "mitohormesis" or "mitohormetic effects".

In line 224, "Calc-calmodulin" may be corrected to "Ca2+/calmodulin".

In line 235, "dimerization of p50 to p65" may be corrected to "dimerization of p50 and p65." It may be worth adding earlier that NF-κB is composed of homodimers/heterodimers of five family proteins.

In line 236, "Iκb-α kinase" may be "IκB kinase".

In line 253-256, "stress-1-related activating transcription factor" is unclear (ATF?), and following "C-Junn-terminal kinase" may be "C-Jun N-Terminal Kinase (JNK)" but this is not the transcription factor.

Please add a space between the number and the unit (line 274).

In Fig. 3, "mitophage" may be typing error of "mitophagy". "Crif1 copies deleted" may be "Crif1 deletion".

In line 338-339, "mitochondrial respiratory complex II subunit succinate dehydrogenase B (SDHB) overexpression and" may be removed for readability or an explanation of SDHB overexpression may be added. "NAC" may be replaced to "N-acetylcysteine"

In line 368, "GST-1α" may be "GSTA1".

In line 377, "AUR" may be replaced to "Auranofin".

In line 430, "scavenging agent" may be corrected to "scavenging protein".

The sentence in line 442-443 may be corrected since AMPK, mTOR, and p38 are not the transcription factor.

In line 457, mouse may also be added in the model organisms.

Comments on the Quality of English Language

English editing is needed before publication as described in the Comments and Suggestion for Authors.

Author Response

Dear Editors and Reviewer #2:

We would like to thank you and the reviewers for your comments on the improvement of our manuscript entitled Harnessing Mitochondrial Stress for Health and Disease: Opportunities and Challenges. The revisions are highlighted in the main text and a point-by-point response is provided below.

My Sincerely,

Ye Zeng

Sichuan University

Responds to the reviewer' s comments:

Reviewer #2: In this review article, the authors overviewed the mechanisms by which mitochondrial stress induces both beneficial effects, known as mitohormesis, and harmfull effects that cause diseases and aging. The manuscript is well structured and covers important topics, however there are a number of unclear descriptions that need correction before publication.

1.Since there are typing errors and use of unfamilier words, some of which is preferentially used by AI (e.g. delve, and paramount), the manuscript should be checked by native English speaker or such edditing service.

2.The meaning of the words "toxicant excitation process" (line 45), "mitochondiral excitatory effects" (line 71), and "mitochondrial excitation" (line 75) are unclear and they should be expressed differently or defined clearly.

Response to Q1 and Q2: Lines 22: “delve” is corrected to “explores”; Line 41: “paramount” is corrected to “important”; Lines 54: “toxicant excitation process” is corrected to “severe oxidative stress and damage cells”; Line 82: “mitochondrial excitatory effects” is corrected to “mitohormesis”; Line 86: “mitochondrial excitation” is corrected to “mitohormesis”.

  1. Although the authors only mentioned "integrated stress response" (line 254), the ISR pathway should also be explaind sufficiently since DELE1-HRI mediated ISR is one of important retrograde pathway elicited by mitochondrial dysfunction. DELE1 is cleaved by OMA1, an upsteam of OPA1 (line 403), and FGF21 and GDF15 (line 399) are known as downstream target genes of ISR.

Response to Q3: Thank you for the nice comments. We added integrated stress response (ISR) in section 3.2.2. Lines 338-345: “In addition, integrated stress response (ISR) is also thought to be an important factor in UPR mt activation. ISR is a retrograde signaling pathway that integrates different stimuli from many organelles, including mitochondria, the endoplasmic reticulum (ER), and the cytoplasm[78]. Under diverse stress conditions, such as maDNA defect, misfolded proteins, mitonuclear imbalance and ROS increase, ISR inhibits general protein translation while favoring the translation of a small subset of mRNAs by activating α-subunit-subunit of eukaryotic initiation factor 2 (eIF2α), as well as  selective translation transcription factors such as ATF4, ATF5, and CHOP [79].”. We also added ISR in section 2. Lines 139-141: “Mitochondrial stress can activate the integrated stress response (ISR), which is also thought to be an important factor in the activation of UPRmt.”

  1. The words "synthesizes" (line 16), "paramount" (line 32), and "prompt" (line 66) should be changed appropriately.

Response to Q4: Line 26: “Based on the available evidence, we elucidate how mitohormesis maintains mitochondrial homeostasis through mechanisms like mitochondrial quality control and mitophagy.”; Line 41: “paramount” is corrected to “important”; Line 77: “prompt” is corrected to “induce”.

5.Please check for grammatical errors. In line 30, "mitochondrial" may be "mitochondria". In line 105, "protein toxicity signals" may be "protein toxic signals" or "protein toxicities".

Response to Q5: Line 39: “mitochondrial” is corrected to “mitochondrial”; Line 129: “protein toxicity signals” is corrected to “protein toxic signals”.

6.Since it is hard to distinguish NRF2 and Nrf2, that stand for Nuclear respiratory factor 2 and Nuclear factor erythroid 2-related factor 2, respectively, tha latter can be abbreviated as NFE2L2.

Response to Q6: All Nrf2 in this article has been modified to NF2L2.

  1. Please check all abbreviations are spelled out when they appeared in the first time in the abstract and in the main text, especially "mtDNA", "PGC-1α", "PTEN", "GSH", "MAPK", "SOD", and "HSF1".

8.Since full spell of some abbreviations are uncommon or wrong (e.g. "NF-κB", "SERCA", and "NFAT"), all abbreviations should also be checked using database such as GeneCards.

9.Common terms such as "NAD+" may not be spelled out.

10.Some abbreviations not appear in the later part may be removed such as "ELF EMF" (line 178), "EC" and "AS" (line 318).

Response to Q7-Q10:  All abbreviations have been checked; Line 358: “SERCA” is corrected to “SERCA1a”; Line 223: “ELF EMF” is removed; Lines 376-378: “EC” and “AS” are removed.

  1. In line 98, "cross-talk factors" is too broad in this context and should be corrected.

Response to Q11: Lines 109-122: “Mitochondrial stress can transport signaling molecules to the nucleus, directly or indirectly regulate the expression of nuclear genes, and trigger adaptive stress responses[13]. When nutrients are abundant, mitochondrial acetyl-coenzyme A (acetyl-CoA) is delivered to the nucleus for intracellular histone acetylation, promoting cell growth and proliferation[14]. Energy deficiency including electron transport chain (ETC) dysfunction, reduces nucleoplasmic acetyl-CoA levels, limits histone acetylation, and induces autophagy[15]. Mitochondrial stress generates ROS. ROS can activate ROS sensors, triggering transcription-activating signaling cascades through redox-dependent modifications of proteins[16]. The ROS-activating signaling pathways lead to the expression of genes that mediate the adaptive response in the nucleus[17]. Ca2+ is released from the mitochondria, which activates transcription factors and improves calcium homeostasis and mitochondrial metabolism via Ca2+ sensitive protein kinases, phosphatases, and transcriptional suppressors[18].”

These are added in the section 2 now.

12.In line 120, the meaning of "pressure" is unclear.

Response to Q12: Lines 145-147:  Pressure is revised to stress now.

13.Since TRPC1 (line 177) is known to contribute to SOCE (line 170), the explanation of this paragraph may be improved.

Response to Q13: Lines 206-225: “ Mitochondria respond to mechanical and electromagnetic stimuli, contributing to cellular signaling pathways involved in oxidative stress regulation. These processes correlated with the expression levels of calcium permeable typical transient receptor potential Channel Type 1 (TRPC1)[38]. Mechanical and electromagnetic cellular responses influence calcium signaling pathways, such as the storage-operated calcium entry (SOCE) pathway, which modulates mitogen-activated protein kinase (MAPK) signaling[39]. These responses vary depending on cell type and environmental conditions, impacting cellular survival and metabolism. TRPC1-mediated calcium entry activates the calcineurin/NFAT pathway [40, 41], maintaining TRPC1 transcription[42], as well as supporting PGC-1α function and oxidative-muscle maintenance[43]. PGC-1α activity is regulated after translation by AMP-activated protein kinase (AMPK) phosphorylation (stimulated by a high AMP/ATP ratio) and NAD1-dependent deacetylase Sirtuin-1 (Sirt1) deacetylation (stimulated by a high NAD+ /NADH ratio)[44, 45]. In healthy muscle cells, low levels of magnetically elevated oxidative stress induce beneficial adaptive developmental responses[39], whereas in metabolically inflammatory breast cancer cells, increased levels of magnetically induced oxidative stress disrupt cell survival but keep healthy cells unharmed[46]. However, the expression of TRPC1 differed in extremely low-frequency electromagnetic field amplitude (1.5mT[39] and 3mT[47], respectively) and exposure time (10 min[39] and 60 min[47], respectively).”

These are revised in section 3.1.2. now.

  1. In line 194, "metabolism" may be "metabolic disorder" or "metabolic stress" because basal metabolism may not be stimulus.

Response to Q14: Line 261: “metabolism” is corrected to “metabolic disorder”.

15.In line 203, "mitotic effects" is unclear and not mentioned in the reference. This may be typing error of "mitohormesis" or "mitohormetic effects".

Response to Q15: Line 270: “mitotic effects” is corrected to “mitohormesis”.

16.In line 224, "Calc-calmodulin" may be corrected to "Ca2+/calmodulin".

Response to Q16: Line 297: "Calc-calmodulin" is corrected to “Ca2+/calcineurin”.

  1. In line 235, "dimerization of p50 to p65" may be corrected to "dimerization of p50 and p65." It may be worth adding earlier that NF-κB is composed of homodimers/heterodimers of five family proteins.

Response to Q17: Line 311: “dimerization of p50 to p65” is corrected to “dimerization of p50 and p65”; We added the content of NF-κB. Lines 305-309: “NF-κB family is composed of five related transcription factors (p65, RelB, c-Rel, p52, and p50), all of which are expressed in skeletal muscle. These transcription factors share an N-terminal DNA-binding/dimerization domain, known as the Rel homology domain, through which they can form homo- and heterodimers[71].”

  1. In line 236, "Iκb-α kinase" may be "IκB kinase".

Response to Q18: Line 312: “Iκb-α kinase” is corrected to “IκB kinase”.

  1. In line 253-256, "stress-1-related activating transcription factor" is unclear (ATF?), and following "C-Junn-terminal kinase" may be "C-Jun N-Terminal Kinase (JNK)" but this is not the transcription factor.

Response to Q19: Line 330: “Mitochondrial import efficiency of the transcription factor ATFS-1 in C. elegans and potentially orthologous transcription factors in mammals (ATF4, ATF5, CHOP) play critical roles in UPRmt activation, facilitating cellular adaptation and mitochondrial homeostasis[20]. UPRmt activate transcription factors, such as C/EBP homologous protein (CHOP) and CCAAT enhancer binding protein beta (C/EBPß), that in turn increase the expression of UPRmt genes. Activation of the UPRmt pathway enhances the adaptive stress response and helps maintain homeostasis by inducing the transcription of mitochondrial molecular chaperones and proteases[77].”

20.Please add a space between the number and the unit (line 274).

Response to Q20: We add a space between the number and the unit. Line 363: “(characterized by elevated resting calcium levels to 100-300 nM) consolidation.”

21.In Fig. 3, "mitophage" may be typing error of "mitophagy". "Crif1 copies deleted" may be "Crif1 deletion".

  1. In line 368, "GST-1α" may be "GSTA1".
  2. In line 377, "AUR" may be replaced to "Auranofin".

24.In line 430, "scavenging agent" may be corrected to "scavenging protein".

25.The sentence in line 442-443 may be corrected since AMPK, mTOR, and p38 are not the transcription factor.

Response to Q21-Q25: We removed the section of mitochondria-related diseases and instead predicted the development of mitohormesis in promoting health. Lines 371-426: “In recent years, the relationship between mitochondrial dysfunction and various diseases has garnered significant attention, particularly in the context of atherosclerosis, cancer, metabolic diseases, aging, and neurodegenerative diseases (Figure 4).

Figure 4. Mitohormetic responses are involved in the initiation and progression of a wide array of disease processes. Mitochondrial dysfunction is widely involved in the occurrence and development of atherosclerosis, cancer, aging and neurodegeneration.

Mitochondrial dysfunction contributes to atherosclerosis by influencing plaque stability[93, 94]. Early stages of atherosclerosis involve mitochondrial DNA (mtDNA) damage, increased ROS production, and progressive respiratory chain dysfunction, leading to endothelial cell dysfunction and plaque vulnerability. Oxidative stress induces autophagy, which, when excessive, can exacerbate cell death and plaque instability[95, 96]. Physiological shear stress maintains vascular endothelial cell mitochondrial homeostasis, while low shear stress-induced inflammation contributes to atherosclerosis progression. Loss of TET2 leads to upregulated expression and activity of mitochondrial respiratory complex II subunit succinate dehydrogenase B (SDHB), disrupting mitochondrial homeostasis, resulting in ROS overproduction and endothelial cell dysfunction[97]. The ROS scavenger NAC can reduce SDHB overexpression and TET2 shRNA induced pyroptosis of vascular endothelial cells[98].

In cancer, apoptosis is a crucial mechanism for inhibiting tumor growth and eliminating cancer cells[99]. By increasing metabolic activity, apoptosis raises levels of ROS within cancer cells, effectively hindering tumor progression. However, excessive ROS can disrupt mitochondrial function, leading to mitochondrial membrane depolarization and the opening of the mitochondrial permeability transition pore (mPTP). This, in turn, activates the intrinsic apoptotic pathway, ultimately resulting in apoptosis and necrosis through the activation of caspases and RIPK1/RIPK3, respectively[100]. Targeting mitochondrial pathways may offer promising strategies for cancer treatment, low-grade mitochondrial stress and the adaptive stress response can mitigate the progression of diseases. During obesity, characterized by excessive nutrient supply, mitochondrial reactive oxygen species (ROS) production increases, prompting mitochondria to enhance biogenesis and antioxidant defense mechanisms[101]. Targeting mitochondrial pathways involved in metabolic regulation may hold promise for treating metabolic diseases and improving overall health.

Aging and neurodegenerative diseases are closely associated with mitochondrial dysfunction and increased production of ROS. Oxidative stress resulting from mitochondrial dysfunction is a key event in the pathogenesis of Parkinson's disease (PD)[102]. In a variety of neurodegenerative diseases, the NF2L2 antioxidant pathway is disrupted, mitochondrial metabolism and ROS homeostasis are significantly impaired, and the level of PGC-1α is significantly reduced[103]. Mild inhibition of mitochondrial function has been shown to prolong lifespan. UPRmt genes are upregulated in Alzheimer's disease (AD), PD, and other neurodegenerative diseases, suggesting a protective role against protein aggregation interfering with mitochondrial function[104, 105].Studies have shown that compounds like chrysanthemum and apigenin can inhibit mitochondrial respiration temporarily, induce early ROS production, and activate the AMPK/NRF-2/FOXO pathway, thereby promoting mitosis, enhancing oxidative stress capacity and cellular metabolic adaptation, and extending the lifespan of C. elegans[106]. These compounds delay aging and improve age-related diseases by targeting mitochondrial function, suggesting potential therapeutic strategies for mitigating neurodegeneration and aging-related decline.

Overall, current research only recognizes severe or irreversible mitochondrial damage in diseases such as atherosclerosis, cancer, metabolic diseases, aging, and neurodegenera-tive diseases, but the early stages of mitochondrial damage and its relationship to disease progression remain unclear. Focusing on early mitochondrial damage may allow for a more accurate assessment of suboptimal health conditions, and holds promising for pro-moting the health by triggering mitohormesis. ”.

These are added in the Section 4 now.

  1. In line 457, mouse may also be added in the model organisms.

Response to Q26: “mouse” is added in Line 433.

Round 2

Reviewer 1 Report

Comments and Suggestions for Authors

The authors did a nice job of responding to my critique.  The revised version reads more fluidily and does provide a comprehensive and exhaustive review of mitohormeis and its molecular mechanisms.  The updated schematics provide clarity and illustrate the complicated pathways described in the narrative.  Finally, the the summary paragraph on mitohormesis and disease provides a nice overview without diverging from the main focus of the review.  

Author Response

Thank you again for your help in improving our manuscript.

Reviewer 2 Report

Comments and Suggestions for Authors

The revised manuscript has been significantly improved and ready for publication after some minor points and typos described below are addressed.

In line 158, "antioxidant medium" may be changed to just "antioxidant" or "antioxidant molecule".

In line 161, gleek fonts of two betas should be corrected.

In line 159, "2" of superoxide anion (O2-) may be subscript (O2). Sometimes an electron of radical is expressed by a dot like "O2・−".

In line 189, "of p66Shc and its translocation to mitochondria" may be added after "Cys-59 residue".

In line 210-211, "AMPK phosphorylation" may be misleading and the sentence may be changed to "PGC-1α activity is regulated by post-translational phosphorylation by AMPK".

In line 212, "NAD1" may by typing error of "NAD+"

In line 284-286, "ULK1" and "AMPK" should be categorized in "protein kinases", and "HIF" should be categorized in "multiple transcription factors".

In line 333, "maDNA" may be "mtDNA".

In line 335, "subunit-subunit" should be corrected.

In line 336, "of" may be added between "selective translation" and "transcription factor".

Comments on the Quality of English Language

Minor points to be corrected are listed above comment.

Author Response

Thank you for all your comments. All are ddressed.

We have also carefully checked and revised the text throughout.